# Systematic analysis of ChatGPT, Google search and Llama 2 for clinical decision support tasks

Sarah Sandmann [1], Sarah Riepenhausen [1], Lucas Plagwitz [1] & Julian Varghese [1]✉

It is likely that individuals are turning to Large Language Models (LLMs) to seek health advice, much like searching for diagnoses on Google. We evaluate clinical accuracy of GPT-3·5 and GPT-4 for suggesting initial diagnosis, examination steps and treatment of 110 medical cases across diverse clinical disciplines. Moreover, two model configurations of the Llama 2 open source LLMs are assessed in a sub-study. For benchmarking the diagnostic task, we conduct a naïve Google search for comparison. Overall, GPT-4 performed best with superior performances over GPT-3·5 considering diagnosis and examination and superior performance over Google for diagnosis. Except for treatment, better performance on frequent vs rare diseases is evident for all three approaches. The sub-study indicates slightly lower performances for Llama models. In conclusion, the commercial LLMs show growing potential for medical question answering in two successive major releases. However, some weaknesses underscore the need for robust and regulated AI models in health care. Open source LLMs can be a viable option to address specific needs regarding data privacy and transparency of training.

The company OpenAI has achieved significant advancements in the perception of Artificial Intelligence (AI) by using Large Language Models (LLMs) in broad segments of the population with the introduction of ChatGPT with its initial version GPT3·5 and its recent release GPT-4[1]. In many areas, LLMs show remarkable potential, particularly in handling pure text tasks. This includes summarizing, rephrasing, and generating novel textual content or programming code[2,3]. Moreover, they enable the user to consult the system as a personal assistant, adept at addressing a wide range of inquiries. However, the nature of existing text corpora used to train such AI systems can contain inconsistencies, incompleteness or information bias[4,5]. Therefore, ChatGPT is frequently criticized for spreading false information with persuasive rhetoric and artificial hallucination[6,7]. This is particularly crucial in fields such as medicine, when confused patients attempt to associate symptoms with diagnoses and therapies. Although these models are not primarily designed for medical consultation, their use can become as commonplace as googling symptoms by both

patients[8,9] and physicians[10,11]. Moreover, preliminary studies have already demonstrated significant potential in utilizing ChatGPT within the medical field. Research has shown that ChatGPT can pass the United States Medical Licensing Exam (USMLE) or Advanced Cardiovascular Life Support (ACLS) exam[12,13]. In addition to exam simulations, previous works have also illustrated the potential benefit of ChatGPT in everyday medicine, for instance, in medical writing by extracting information from electronic health records, assisting with literature searches or providing guidance on writing style and formatting[14]. Furthermore, entire specialized fields have discussed the use of LLMs, such as enhancing services in dental telemedicine or by improving patient-centered care in radiology[15,16]. The diagnostic capability of GPT-3·5 has also been demonstrated in its basic aspects, as it can generate a well-differentiated diagnosis list for common chief complaints[17]. We aim to extend upon research of this nature through our comprehensive analysis. To the best of our knowledge, there has been no prior work on thorough evaluation encompassing various

[1]Institute of Medical Informatics, University of Münster, Münster, Germany. ✉e-mail: julian.varghese@uni-muenster.de

clinical decision-support tasks, including diagnostic and therapeutic capabilities.

We aim at investigating clinical accuracy of two well-established successive LLMs – GPT-3·5 and GPT-4 - regarding three key tasks of clinical decision-making: 1) initial diagnosis, 2) examination and 3) treatment.

For each of these three tasks, the influence of a disease's frequency on performance is additionally evaluated (categories rare, less frequent and frequent). It is known that machine learning models perform poorly on problems with little or no training data available. Based on the known incidence and prevalence rates of the diseases, we consider our assessment is apt to provide different levels of difficulties to the AI. In particular, patients with rare diseases run a higher risk of being undiagnosed, yet collectively, they constitute a significant portion of the population[18]. These patients are in need of diagnostic support and are likely to utilize AI tools more frequently.

As input for the assessment, we consider clinical reports representing a broad disease entity spectrum. The case reports were extracted, processed to first-person perspective in layman language and translated from German clinical casebooks. Two major publishers requiring licensed access were taken into account. The restricted access, the unavailability in English, and the subsequent conversion into layman terms collectively serve as safeguards in order to minimize the likelihood that ChatGPT was trained on this input.

For initial diagnosis, we benchmark the performances against the Google search engine. In a sub-study we additionally explore the potential of open source models at the example of Llama 2, a family of LLMs that have recently outperformed state of the art open source models in general chat model tasks[19].

In our in-depth analyses, we present a comprehensive evaluation of the current performance trajectory of two successive LLM versions with respect to their capacity to facilitate diagnostic decision-making, recommend appropriate diagnostic examination procedures, and propose treatment options across a wide spectrum of diseases, including those that are rare, less frequent, and frequent.

## Results

### Inter-rater reliability

Regarding diagnosis, highest levels of agreement can be observed with $\kappa = 0.8$ for GPT-3·5, $\kappa = 0.76$ for GPT-4 and $\kappa = 0.84$ for Google. Examination is characterized by $\kappa = 0.53$ for GPT-3·5 and $0.64$ for GPT-4. With respect to treatment, we observe $\kappa = 0.67$ for GPT-3·5 and $\kappa = 0.73$ for GPT-4. According to Landis et al. 1977, this corresponds to substantial (0·61–0·8) to almost perfect (0·81–1) agreement[20]. The findings do not provide any evidence of consistent rating discrepancies favoring one rater over the other. Details of results are provided in Supplementary Fig. 2 and Supplementary Data 1.

Considering the subset of $n = 18$ cases with additional evaluation of Ll2-7B and Ll2-70Bb, comparable results can be observed (Supplementary Fig. 3 and Supplementary Data 1). There is substantial to almost perfect agreement, no systematic rating discrepancies appear evident.

### Performance evaluation of GPT-3·5, GPT-4 and Google

Figure 1 sums up the results of pairwise comparisons (bubble plots) as well as performance for each of the three disease frequency subgroups (cumulative frequency plots). Performance distributions of all models for different tasks are summarized with bar plots in Supplementary Fig. 4, and violin plots in Supplementary Fig. 5. Information on the median performance and 95% confidence intervals is provided in Supplementary Table 1, p-values and adjusted p-values for all tests performed in Supplementary Table 2.

With respect to diagnosis, all three tools were evaluated. Pairwise comparison showed a significantly better performance of GPT-4 (median: 4·5, IQR = [3·81;4·75]) against both GPT-3·5 (median: 4·25, IQR = [3·0;4·75], $p = 0.0033$) as well as Google (median: 4·0, IQR = [2·75;4·75], $p = 0.0006$). However, no significant difference between GPT-3·5 and Google was observed ($p = 0.6215$). Considering disease frequency, plots in Fig. 1a indicate a continuously better performance for frequent compared to rare diseases. This observation was made for all tools (dark blue line – frequent – rising steeper compared to light blue line – rare). GPT-3·5 performed significantly better on frequent compared to rare diseases ($p < 0.0001$), while GPT-4 showed significant results for frequent vs rare ($p = 0.0003$) as well as for less frequent vs rare ($p = 0.0067$). For Google, no differences between rare and less frequent diseases were observed (Fig. 1a). Despite some differences being visible compared to frequent diseases, results are not significant.

Considering examinations, we compared GPT-4 (median: 4·5, IQR = [4·0;4·75]) to GPT-3·5 (median: 4·25, IQR = [3·75;4·5]). Pairwise comparison showed superior performance of GPT-4 ($p < 0.0001$). Evaluating performance of the two models in relation to disease frequency, results indicated superior performance of GPT-3·5 with respect to frequent diseases. However, these results were not significant. GPT-4 showed comparable performance for both frequent and less frequent diseases, but significantly better performance compared to rare ($p = 0.0203$).

Regarding treatment options, comparing performance of GPT-4 (median, 4·5, IQR = [4·0;4·75]) to GPT-3·5 (median: 4·25 (IQR = [4·0;4·69]) fewer differences were observed. Results in Fig. 1c indicated superior, however, not significant performance of GPT-4 ($p = 0.0503$). Any influence of the diseases' frequency on performance could not be observed.

### Comparison to open source model

Figure 2 visualizes the performance of GPT-3·5 and GPT-4 with violin plots considering all 110 cases and dots highlighting performance of the 18 selected cases in comparison to Llama-2-7b-chat (Ll2-7B) and Llama-2-70b-chat (Ll2-70B). The performance of all models separated by disease frequency is provided in Supplementary Fig. 6.

The median scores and interquartile ranges of the nine GPT-4 top-scoring cases were 14·5 [14·5;14·75], 14·0 [13·25;14·0], 12·25 [11·25;13·5] and 11·75 [11·25;12·75] for GPT-4, GPT-3·5, Ll2-7B and Ll2-70B respectively. Analogously, for the nine worst-scoring cases: 11·0 [9·25;11·25], 10·25 [9·5;10·5], 10·25 [8·5;11·0] and 8·5 [7·75;10·25].

Overall, we observe slightly inferior performances of the open-source LLMs compared to GPT3·5 and GPT-4. Additionally, we cannot observe a noticeable performance difference between the two open-source LLM configurations.

## Discussion

Our study presents results from a systematic evaluation of different clinical decision support tasks with the principal models GPT-3·5, GPT-4 and naïve Google search in a broad spectrum of diseases entities and disease incidences. The results indicated a clear performance progression of GPT-4 compared to its predecessor GPT-3·5 and Google search. In the sub-study, both the simple 7b and the more complex 70b open source Llama 2 model showed slightly inferior performance with considerable variation. Given that these are not continuously updated as their commercialized ChatGPT counterparts, which might benefit from greater financial resources, we view these results as quite remarkable. Moreover, the possibility to deploy open source models locally and to fine-tune them towards user-specific needs could not only improve performances but also significantly mitigate concerns regarding data protection and lack of transparency in model training.

All of the three principal models performed worst in the most difficult task – the initial diagnosis recommendation. Despite the high median performances of the best performing model GPT-4, the full score of 5/5 points was reached only in 18, 24, and 26 out of 110 cases for the tasks diagnosis, examination and treatment respectively.

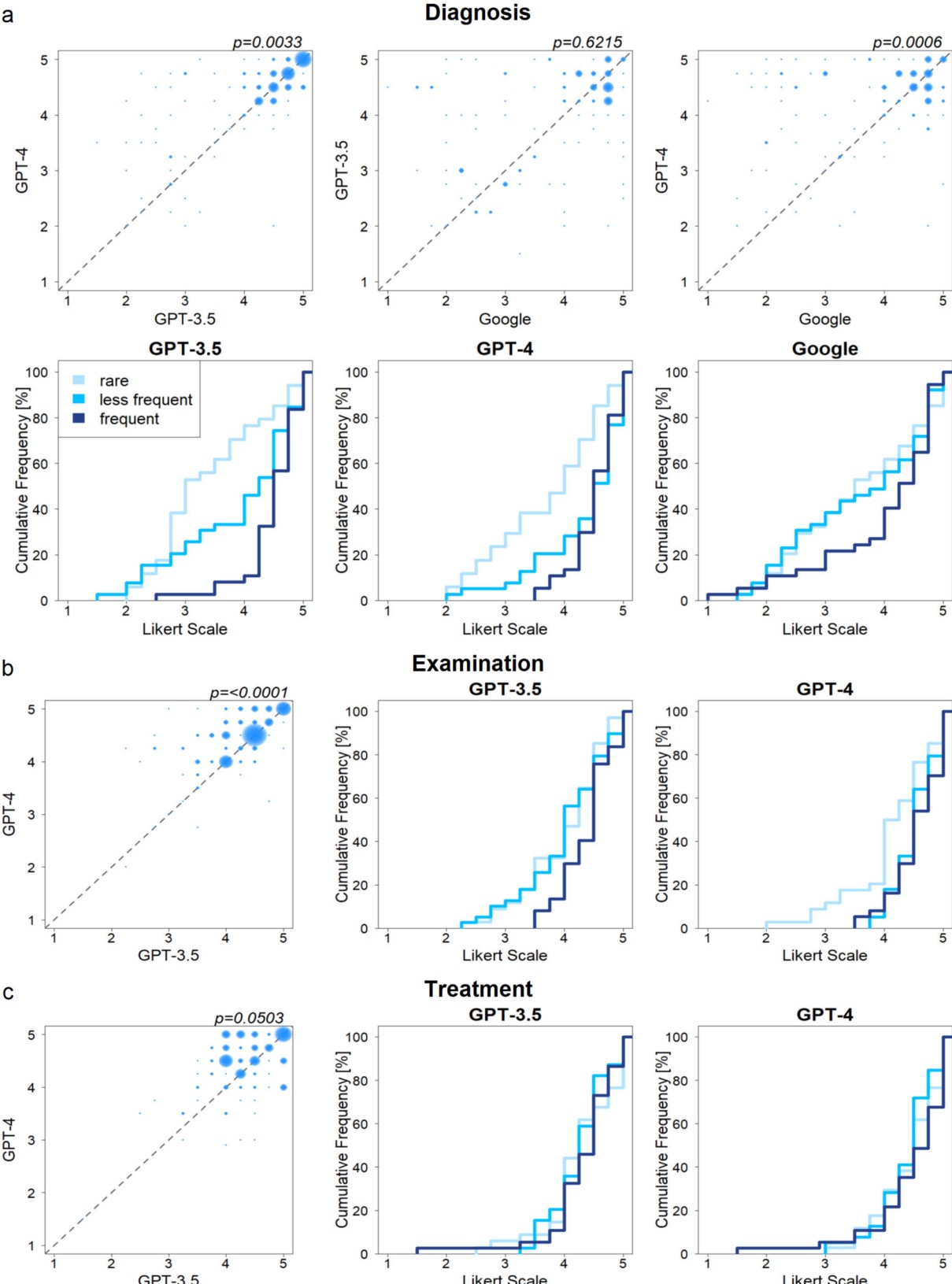

**Fig. 1 | Performance comparison of GPT-3·5 vs GPT-4 vs Google. a** Performance of GPT-3·5 vs GPT-4 vs Google for diagnosis. **b** Performance of GPT-3·5 vs GPT-4 for examination (exact adjusted *p*-value *p* = 3.2241·10$^{-6}$). **c** Performance of GPT-3·5 vs GPT-4 for treatment. Bubble plots show the pairwise comparison of two approaches. Cumulative frequency plots show the cumulative number of cases (Y-axis) and their accuracy scores (X-axis) for each disease frequency subgroup (light blue: rare, intermediate blue: less frequent, dark blue: frequent). One-sided Mann-Whitney test was applied for statistical testing (adjusted with Bonferroni correction for multiple testing considering *n* = 12 tests for diagnosis, *n* = 7 tests for examination and treatment).

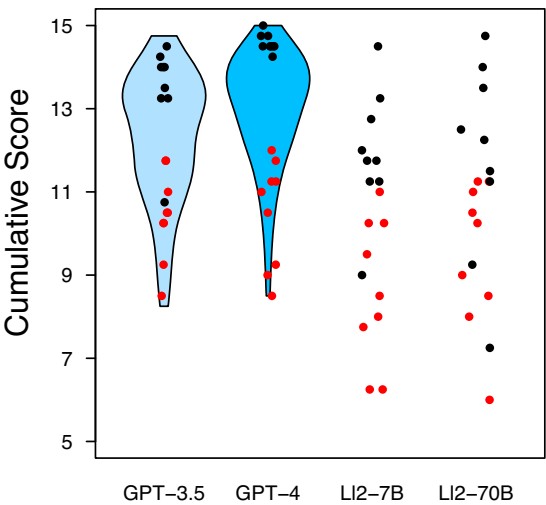

**Fig. 2 | Performance comparison of GPT-3·5 vs GPT-4 vs Ll2-7B vs Ll2-70B considering top-3 and bottom-3 cases.** Black dots mark the top-3 cases based on GPT-4's cumulative score for rare, less frequent and frequent diseases. Red dots the bottom-3 cases. Violin plots visualize the performance of GPT-3·5 and GPT-4 for all $n = 110$ cases. Ll2-7B: Llama-2-7b-chat; LL2-70B: Llama-2-70b-chat.

All tools performed worse in rare diseases compared to frequent ones, which was anticipated as rare diseases are under-represented in large text corpora on the internet.

From a technical perspective, the results are promising as GPT-3·5 and GPT-4 were never trained particularly for such clinical decision support tasks and perform in a considerable amount of cases significantly better than naïve Google search while saving search time and utilizing a more intuitive chatbot functionality to users.

From a clinical point of view, we have to conclude that there are few but still a noticeable number of cases, in which all of the two models could not reach the highest score. This could lead to misunderstandings or medical issues if patients rely on these outputs without contacting trained health professionals. To illustrate an example, there was one case with a patient having pheochromocytoma, a rare neuroendocrine tumor that causes hypertension. The models have correctly picked alpha-adrenergic and beta-adrenergic blockade as potential medication treatment. However, they failed to emphasize that the beta-adrenergic blocker should never be applied first, because the blockade of vasodilatory peripheral beta-adrenergic receptors with unopposed alpha-adrenergic receptor stimulation can further increase blood pressure[21]. A post-hoc test was conducted by re-entering the treatment question after finalizing our results, which generated an improved answer with correct ordering of both agent though we used the same model versions. This occurred likely due to continuous model optimizations at the backend. Our results can therefore only present performance snapshots. Another noteworthy example, in which diagnostic assessment remained poor in all LLMs, including the open source models, was a case with Dermatomyositis, a rare condition in which both skin and muscles are affected. Given the patient characteristics including muscle weakness, facial erythema, gender and age, the condition Dermatomyositis was a likely diagnosis, but was missed by all LLMs, also in the post hoc testing. At this observation, we reiterated and refined the prompt "Could there be other diagnoses, particularly matching the symptoms?" after which GPT-4 proposed the right diagnosis. Interestingly, the manual Google search was superior and received almost perfect consensus score in this case. We believe that the condition's rarity is a key issue that may have misled the LLMs. Highly specific but low frequency information are circumstances in which the use of search engines may provide significantly better information retrieval. In general, prompt refinements or prompt engineering can be powerful methods to improve LLM's output.

The large number of medical case reports from different major medical disciplines and different disease frequencies is a strength of this this study. All cases were extracted from non-English, non-open access sources and further processed by non-medical experts. This procedure comes with strengths and limitations. While it reduces the risk of prior training bias, ensures layman comprehension and thus mimics patient-generated input, it is a type of constructed but not original patient input. This should also be considered in relation to the prompt structure. It is widely recognized that prompt engineering can significantly increase the performance of the model's response[22]. In our prompt strategy, we have not specifically emphasized medical answers, aiming to mirror normal usage and ensure a fair comparison with the Google search engine.

The assessment of clinical accuracy was based on a straight-forward 5-point Likert scale for all three tasks. While this approach facilitates a standardized comprehension of assessment among different physicians and different clinical disciplines, it is still a subjective instrument. Different clinical dimensions of accuracy (e.g. quality of life, surrogate parameters, life expectancy, overall patient safety or harm) were not considered separately and will require further research with multi-dimensional rating scales.

The selection of case reports excluded cases in which the diagnosis overly relies on lab-testing or imaging. This approach was necessary to focus on initial diagnostic assessment based on patient-reported history, where non-existence of lab or imaging findings is the usual case. However, many diagnoses will require them and future studies are needed to assess the diagnostic capabilities when gradually adding such information into the user prompts. Noteworthy, the current version of ChatGPT supports automatic imaging-analyses in conjunction with processing free-text queries.

As mentioned, the analyzed LLMs are continuously being updated by their providers. This could lead to slightly different results if our test cases are re-entered, which limits the reproducibility of the exact performance results. Thus, our results need to be interpreted as snapshots of their time. To take this into account, we studied not only one LLM but two different successive versions with major updates (GPT-3·5 vs GPT-4). In this way we could show improved performance by the significantly updated model, which is explainable by technical progressions and updated text corpora – GPT-3·5's training input only goes to 2021. It is foreseeable that future models could be retrained for the most current text input more efficiently. In particular, task-specific medical LLMs could be connected to quality-approved literature databases such as PubMed and UpToDate, will no longer be outdated and will continuously learn timely after newest published research articles are online. As with all clinical decision support tools that qualify the status of a medical device, LLMs will have to face high regulatory requirements and their approval will be significantly more cumbersome in the light of the low transparency of data input and low explainability in the training process[23].

Our study shows promising performance results by current LLMs for clinical decision support tasks. Apart from medico-legal issues that arise in the context of medical devices and data protection regulations[24], at the current state, they should not be used for medical consultation alone as the performance lacks consistent high accuracy. However, the study evaluated noticeable performance progress going from GPT-3·5 to GPT-4 and outperforming naïve Google search in terms of performance and time-consumption.

Future models will likely be capable of efficiently incorporating quality-approved up-to-date knowledge but will still need to face strict regulations regarding data protection and medical devices in order to be used with medical purpose. In this context, open source LLMs can be a viable option, in which more transparent training processes and human oversight can be implemented. Moreover, they have shown

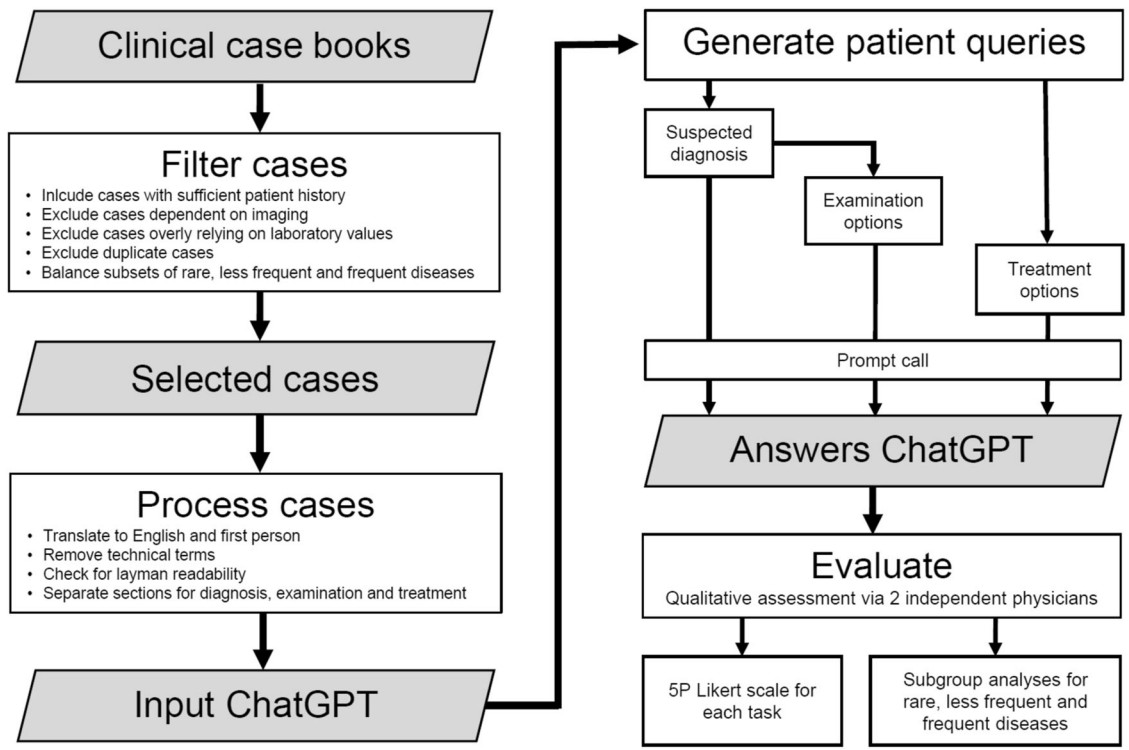

**Fig. 3 | Overview of process steps.** Cases from clinical case books were filtered and processed to generate patient queries for GPT-3·5 and GPT-4. The answers on suspected diagnosis, examination and treatment options were evaluated by two independent physicians and rated on a 5-point Likert scale.

promising preliminary performance in our sub-study. We hypothesize that some of the future AI systems in healthcare will increasingly involve LLMs. Simultaneously, these systems will need to intensify their commitment to adhering to transparency, interpretability and stringent regulatory frameworks pertaining to data security and medical device regulations.

## Methods
An overview of the workflow we applied to select clinical case reports, to query GPT-3·5 and GPT-4 and to evaluate the results is illustrated by the flowchart in Fig. 3.

### Selecting clinical case reports
In order to base our study on a representative, unbiased selection of realistic cases, we screened the case reports published in German casebooks by two publishers: Thieme and Elsevier. Main clinical specialties were internal medicine, surgery, neurology, gynecology and pediatrics. A total of 12 casebooks were utilized, comprising five from Thieme and seven from Elsevier. Among these, five books from each publisher were explicitly titled with the aforementioned medical specialties. Two further casebooks from Elsevier focusing on Rare Diseases and General Medicine were added to further include disease cases of very low incidence and from outpatient settings. This selection generated a pool of 1020 cases.

To study the performance of ChatGPT in relation to the incidence of diseases, cases were categorized in three frequency subgroups as follows: A disease is considered frequent or less frequent if its incidence is higher than 1:1000 or 1:10,000 per year respectively. A disease was considered as rare if the incidence is lower than 1:10,000. Only if no information of incidence was available, the disease was considered rare if the prevalence is lower than 1:2000. Power calculations were conducted to infer a sample size of 33-38 per subgroup in order to achieve a total power of 0·9 (details in Supplementary Methods and Supplementary Fig. 1).

To limit the scope of the subsequent analysis, a random sampling of 40% of the total 1020 cases was performed, ensuring an even distribution of sources, resulting in the examination of 408 cases. This proportion was determined by systematic testing, to find a balance between review effort and statistical power for our subgroup analysis. Initially, a 30% sample resulted in an insufficient number of cases satisfying our following inclusion criterion. Augmenting to 40% ensured each subgroup attained $n \geq 33$ and the total data set reached $n > 100$.

The majority of cases followed a consistent structure of subsequent sections including medical history, current symptoms, examination options and findings, actual diagnosis, differential diagnoses and treatment recommendations. For each of the medical sub-specialties, a physician reviewed all cases and included a case for further analysis when the following criteria were met: (1) the patient or someone else is able to provide information on medical history (e.g. excluding severe traumatic patients), (2) images are not required for diagnostic purposes, (3) diagnosis does not overly rely on laboratory values, (4) the case is not a duplicate. Steps 2 and 3 were necessary as our scope for the diagnosis task was only to assess initial diagnosis, in which a patient does not have any imaging or laboratory findings available. A total of 153 cases were identified as conforming to the established inclusion criterion.

Incidence rates for the selected case studies were researched and sorted into three previously mentioned frequency groups. To ensure a balanced representation, we aimed to include a similar number of cases from each medical specialty while considering both the incidence rates and the publishing source, which resulted in a final selection of 110 cases. Table 1 provides a distribution overview of the included cases, organized by publisher, clinical specialty, and disease frequency. Detailed information including the exact patient medical history is provided in Supplementary Data 1.

Included cases were translated to English by initially using the translation tool Deepl.com, followed by a manual review to ensure

both linguistic accuracy and quality. In order to mimic a true patient situation, cases were processed to patient layman language. Medical history was provided in first-person perspective, containing only general information and avoiding clinical expert terminology. Layman readability was independently checked by two non medical researchers and alternative laymen terms were provided if expert terminology was used. In cases of disagreement for laymen translation, consensus was achieved by a third non medical researcher.

### Querying GPT-3·5 and GPT-4

For generating the patient queries, the analysis plan reads as follows:

I. Open a new ChatGPT conversation.
II. Suspected diagnosis: Write patient medical history and current symptoms, add "What are most likely diagnoses? Name up to five."
III. Examination options: "What are the most important examinations that should be considered in my case? Name up to five."
IV. Open New ChatGPT conservation
V. Treatment options: Write patient medical history and current symptoms and add: "My doctor has diagnosed me with (specific diagnosis X). What are the most appropriate therapies in my case? Name up to five."

All prompts were systematically executed between 3 April 2023 and 19 May 2023 through the website https://www.chat.openai.com. Output generated with GPT-3·5 and GPT-4 for each of the three queries is provided in Supplementary Data 1.

### Querying Google

Symptoms were searched and the most likely diagnosis was determined based on the first 10 hits reported by Google. Search, extraction and interpretation was performed by a non medical expert, mimicking the situation of a patient.

Based on medical history of every case, search strings were defined: baby(opt.) child(opt.) diagnosis < symptoms > previous < previous illness > (opt.) known < known disease > (opt.) < additional information > (opt.).

Symptoms were extracted based on the information provided in medical history. Optionally, if a previously resolved illness existed and

was considered relevant by the non medical expert, the word "previous" was added, followed by information on the disease. If a known condition, e.g., hay fever, existed and was considered relevant, the word "known" was added, followed by information on the disease.

The search was performed in incognito mode using https://www.google.com. For every case, the first 10 websites were evaluated. Websites were scanned for possible diagnoses. The non medical expert compared the symptoms characterizing each diagnosis on the search results to those provided by the medical history. Information on e.g. age or sex – available in the medical history but not by the search string – was additionally taken into account. For example, ectopic pregnancy is not considered possible for a male case with abdominal pain. For cases considering children below the age of 1, "baby" was added to the search string. For cases considering children below the age of 16, "child" was added to the search string.

Solely information available on the websites was evaluated. No further detailed search on a specific diagnosis was performed if, e.g., only limited information on a disease's characteristics was provided by a website.

A maximum of five most likely diagnoses were determined. If more than five diagnoses appear equally likely, the most frequently reported diagnoses were selected. The Google search strings as well as identified five diagnoses are available in Supplementary Data 1.

### Explorative analysis of a further open source Large Language Model

In order to include recently established open source LLMs[19] into our evaluation, we additionally deployed Llama 2 with two different model sizes: Llama-2-7b-chat (Ll2-7B with 7 billion parameters) and Llama-2-70b-chat (Ll2-70B with 70 billion parameters Patient queries were generated analogously to GPT-3·5 and GPT-4: Starting with the system prompt "You are a helpful assistant", followed by a query format consistent with that used for GPT. All queries were formulated with specific parameters: a temperature setting of 0.6, top_p set to 0.9, and a maximum sequence length of 2048. Output generated with the Ll2-7B and Ll2-7Bfor each of the three queries is provided in Supplementary Data 1.

### Performance evaluation

Assessment of the answers generated with GPT-3·5, GPT-4, Google, Ll2-7B, and Ll2-70B was conducted independently by two physicians. The AI's output was reviewed in relation to solutions provided by the casebooks. In cases of uncertainty, further literature was consulted. Each physician scored clinical accuracy based on a 5-point Likert scale according to Table 2. The final score is calculated as the mean of the two individual scores.

All statistical analyses were conducted using R 4.3.1[25]. To assess inter-rater reliability, weighted Cohen's kappa and 95% confidence intervals (CIs) were calculated for each of the three tasks, using the R package DescTools 0.99.54[26]. To explore the performance of GPT-3·5, GPT-4 and Google, an independent evaluation of diagnosis, examination and treatment was conducted. The performance of GPT-3·5 vs GPT-4 was evaluated by applying a paired one-sided Mann-Whitney test (R package base, function wilcoxon.test). Considering diagnosis, paired one-sided Mann-Whitney tests comparing GPT-3·5 vs Google and GPT-4 vs Google were additionally performed. Possible influence

### Table 1 | Overview of clinical cases

|  | All | Rare | Less Frequent | Frequent |
|---|---|---|---|---|
| *n* | 110 | 34 | 39 | 37 |
| Publisher |  |  |  |  |
| • Elsevier | 48 | 16 | 19 | 13 |
| • Thieme | 62 | 18 | 20 | 24 |
| Clinical specialties |  |  |  |  |
| • Gynecology | 19 | 0 | 9 | 10 |
| • Internal Medicine | 28 | 9 | 11 | 8 |
| • Neurology | 20 | 10 | 5 | 5 |
| • Pediatrics | 23 | 8 | 9 | 6 |
| • Surgery | 20 | 7 | 5 | 8 |

Number of cases per publisher and clinical specialties, considering all, rare, less frequent and frequent diseases.

### Table 2 | Assessment scheme

| 1 | 2 | 3 | 4 | 5 |
|---|---|---|---|---|
| Most of all relevant options were not mentioned. All or most of the system's generated options were redundant or unjustified. | Some or many relevant options were not mentioned. Some of the system's generated options were redundant or unjustified. | Most of all relevant options were mentioned. Some of the system's generated options were redundant or unjustified. | Most of all relevant options were mentioned. Few of the system's generated options were redundant or unjustified. | All of the relevant options were mentioned. There was no redundant or unjustified option mentioned. |

Definition of the assessment scheme for diagnosis, examination and treatment recommendations.

of the incidence (rare vs less frequent vs frequent) was investigated using non-paired one-sided Mann-Whitney test. With Bonferroni correction[27], a conservative adjustment for multiple testing was applied (for examination and treatment: 1 test comparing GPT-3·5 to GPT-4, 3 tests within group GPT-3·5, and 3 tests within group GPT-4 resulting in $n = 7$; for diagnosis: 3 tests comparing GPT-3·5 to GPT-4 to Google, 3 tests within group GPT-3·5, 3 tests within group GPT-4, and 3 tests within group Google resulting in $n = 12$).

The explorative evaluation of Ll2-7B and Ll2-70B was performed as an additional sub-study after new developments and promising performances of open source LLMs were recently reported, specifically regarding the Llama 2 family[19]. As the main focus of our work is to investigate the progress of LLMs at the example of ChatGPT, we powered our study for the above mentioned 7 and 12 comparisons respectively. Investigating the performance of Ll2-7B and Ll2-70B solely serves to provide a descriptive analysis. To compare the overall performances, we calculated the cumulative score over all three tasks – diagnosis, examination and treatment. The score was sorted according to performance of GPT-4, separately for the three subgroups rare, less frequent and frequent diseases. For each of the subgroups, the top-3 and bottom-3 cases were chosen, resulting in a subset of $n = 18$ out of 110 cases. In case of ties, the corresponding case was chosen randomly.

## Reporting summary

Further information on research design is available in the Nature Portfolio Reporting Summary linked to this article.

## Data availability

Data on all clinical cases, output from GPT-3·5, GPT-4, Google, Ll2-7B, and Ll2-70B and their assessment are provided in the Supplementary Data 1.

## Code availability

The R scripts generating Figs. 1 and 2 of the main manuscript are available as Supplementary Data 2 and 3. The R script performing sample size calculation is available as Supplementary Data 4.

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

## Acknowledgements

We acknowledge support from the Open Access Publication Fund of the University of Münster (S.S., S.R., L.P., J.V.).

## Author contributions

All authors made substantial contributions to the conception or design of the work; the acquisition, analysis, or interpretation of data; drafting or revising the paper. All authors approved the paper. All authors agreed both to be personally accountable for the author's own contributions and to ensure that questions related to the accuracy or integrity of any part of the work are appropriately investigated, resolved, and the resolution documented in the literature. J.V. and S.S. designed the study. J.V. supervised the study. L.P. and S.R. performed literature research, filtered, and prepared the clinical cases to generate patient queries. L.P. queried and configured GPT-3·5, GPT-4, Google, Ll2-7B and Ll2-70B. S.S. queried Google. S.R. and J.V. evaluated all answers. S.S. performed analysis of the data. The paper was prepared by J.V., S.S., and L.P. with input from all authors.

## Funding

## Competing interests

The authors declare no competing interests.
