## [Peer Review File · Nature Communications]

Systematic Analysis of ChatGPT, Google Search and Llama 2 for Clinical Decision Support TasksREVIEWER COMMENTS

Reviewer #1 (Remarks to the Author):

This paper evaluated the capability of GPT-3.5 and GPT-4 for suggesting diagnoses, examinations, and treatments. Samples were carefully extracted from German clinical casebooks and converted to consumer language. The authors also paid attention to the sampling across diseases from three levels of prevalence and different clinical specialties. Statistical analyses were sound, except for one suspicious case noted below. The language was overall accurate and easy to follow, except for the introduction.

Major concerns in methodology and significance

- My primary concern about the paper's scientific value is that the authors focused only on two closed-source large language models (LLMs), GPT-3.5 and GPT-4. Currently, every few days, there is a new open-source LLM coming out to compete with the others. Although GPT and Claude are currently the best-performing LLMs overall, reviewing the open-source models is essential, not only because they are quickly catching up with the commercial ones but, more importantly, because we know how they are trained. Evaluating how they perform can guide the approach of how to train them. Otherwise, it would be like commercial product reviews. Specifically, I would suggest the authors choose well-performing models from popular LLM leaderboards, such as the Chatbot Arena (<https://huggingface.co/spaces/lmsys/chatbot-arena-leaderboard>) and the Open LLM Leaderboard (https://huggingface.co/spaces/HuggingFaceH4/open_llm_leaderboard). Notable models that people often compare to and should be included for evaluation include e.g., the Llama-2-chat series, the WizardLM series, and the Vicuna series. I suggest the authors compare models of different sizes (7B, 13B, 30B, 70B) from the same series and see how size affects accuracy, instruction following capability, alignment to human preference, and hallucination. Another potential angle is how models trained in different approaches perform this task. For example, there are models trained by human-written questions, automatically generated questions, and those trained in the FLAN fashion. They perform differently in different tasks and should be compared and studied.
- It is known that LLMs are sensitive to how the prompts are phrased, which is why we spend much time on prompt engineering to generate desired outputs stably. The authors

mainly focus on an online situation where a consumer seeks medical consultation. Therefore, the input language is expected to be varied, with some prompts better at querying certain LLMs than others. The comparison between GPT-3.5 and GPT-4 in this paper specifies a common prompt template, which makes the comparison “fair” but not enough to reflect these LLMs’ capability to comprehend various ways of asking medical questions.

Other comments:

- Line 116: It is a good attempt to test GPT-3.5 and GPT-4 with information not seen in their training data. However, restricted data access cannot guarantee it. Tech companies competing in LLMs try every possible way to gain more training data, including scanning printed books and purchasing databases, legal or illegal. There have been multiple lawsuits alleging OpenAI using copyright materials to train GPT-3.5. I have also observed that GPT-3.5 and GPT-4 know very well how doctors write medical records in the Chinese language, which could only mean that they have seen Chinese medical records in their training data, and I could not think of a legal way how they obtained it in the US. Therefore, one should be very conservative when claiming GPT-3.5 or GPT-4 have not seen something in their training data.
- Line 160: The inclusion criterion rules out cases requiring imaging or lab results, which makes the problem much easier than real clinical encounters. A common criticism about automatic diagnosis models from doctors is that they overly rely on symptoms, while in practice, there are not many diseases that can be diagnosed without imaging and lab tests. The authors should at least mention this as a limitation.
- Line 281: A p-value of 1 looks very strange. The lower part of Figure 2.A shows that the distribution of scores is different between GPT-3.5 and Google. Can the authors check if the test direction is correct in their one-sided test? They can swap the direction or use a two-sided test and see if the conclusion holds.
- Line 369: “As mentioned, the analyzed LLMs are continuously being updated through fine-tuning by their providers. This could lead to slightly different results if our test cases are re-entered, which limits the reproducibility of the exact performance results. Thus, our results need to be interpreted as snapshots of their time. To take this into account, we studied not only one LLM but two different successive versions with major updates (GPT-3.5 vs GPT-4).”

The first part of this statement is true: these models are being updated by OpenAI, and the performance changes over time. Thus, one should experiment on different versions of the same model to study their change or stability, e.g., gpt-3.5-turbo-0613 vs. gpt-3.5-turbo-0301, instead of two different models. Also, to improve reproducibility, the authors should specify the model version (e.g., gpt-3.5-turbo-0301) and use a temperature of 0.

- The language in the Introduction needs grammar editing and is not accurate enough. Starting from Methods, it feels much more fluent, accurate, and easy to follow.

Reviewer #2 (Remarks to the Author):

The main contribution of this paper is to investigate the clinical accuracy of two well-established successive LLMs GPT-3.5 and GPT-4, in terms of three key levels of clinical decision-making: diagnosis, examination and treatment, as well as the influence of disease frequency on performance.

The study's objective value adds to the prior studies in evaluating GPT-3.5 and GPT-4 as clinical decision support tools, beyond medical examination question answering and administrative tasks, to gain insight on diagnostic decision-making, recommend appropriate diagnostic procedures, and propose treatment options for a wide range of diseases.

In terms of methodology:

A1. The study took good efforts to ensure test dataset was representative of realistic clinical cases and minimized bias. The dataset comprised case reports extracted from German clinical casebooks, across the major specialities in medicine (internal medicine, surgery, neurology, gynecology and pediatrics). The restricted access, unavailability in English, and subsequent conversion into layman terms, were safeguards taken by the authors to minimize the chance of ChatGPT being trained on the testing dataset. A decently sizeable number of cases (110 total cases) were included in the analysis.

A2. The workflow in extracting the final number of cases for the test dataset was clear and detailed. Of which, how was the random sampling of 40% of the total number of cases

decided upon?

A3. Were there copyright issues breached in entering the content from non-open access clinical casebooks into GPT (which may be used in GPT model training)?

A4. Good choice of statistical analyses were selected for the evaluation of responses.

In terms of results:

B1. For the cumulative frequency plots, it is less interpretable for the reader to look at the proportion of responses scored as each point on the Likert scale, for each tool.

B2. More sub-analyses such as the pheochromocytoma case would be insightful, as there is a limit to the analysis of responses using the 5-point Likert scale based on 1 criteria only (clinical accuracy). Perhaps examples of 'dangerously erroneous' information which may cause harm to patients, if present, can also be included.

Rebuttal Letter

In the following, we present our point-by-point response. Our responses are highlighted in green.

Reviewer #1

Major concerns in methodology and significance

- My primary concern about the paper's scientific value is that the authors focused only on two closed-source large language models (LLMs), GPT-3.5 and GPT-4. Currently, every few days, there is a new open-source LLM coming out to compete with the others. Although GPT and Claude are currently the best-performing LLMs overall, reviewing the open-source models is essential, not only because they are quickly catching up with the commercial ones but, more importantly, because we know how they are trained. Evaluating how they perform can guide the approach of how to train them. Otherwise, it would be like commercial product reviews. Specifically, I would suggest the authors choose well-performing models from popular LLM leaderboards, such as the Chatbot Arena (<https://huggingface.co/spaces/lmsys/chatbot-arena-leaderboard>) and the Open LLM Leaderboard (https://huggingface.co/spaces/HuggingFaceH4/open_llm_leaderboard). Notable models that people often compare to and should be included for evaluation include e.g., the Llama-2-chat series, the WizardLM series, and the Vicuna series. I suggest the authors compare models of different sizes (7B, 13B, 30B, 70B) from the same series and see how size affects accuracy, instruction following capability, alignment to human preference, and hallucination. Another potential angle is how models trained in different approaches perform this task. For example, there are models trained by human-written questions, automatically generated questions, and those trained in the FLAN fashion. They perform differently in different tasks and should be compared and studied.

We are more than happy to address your primary concern, as it would provide a good comparison with open source models and thus would raise more interest of our manuscript for the technical NLP community in medicine. Before writing how we address it, we also would like to add some challenges that come with new comparisons, especially from a statistical point of view.

- Our initial scope, as delineated in the title, aims to examine the progression between GPT-3.5 and GPT-4. We believe our focus should remain here, as these models' accessibility through user-friendly interfaces significantly impacts their current real-world application by laymen patients who might not be able to deploy and run open-source models. But we also see the potential that usability and accessibility of open-source models could match up in the longer term.
- Our current evaluation was rigorously preceded by a power analysis that determined the number of cases required in order to achieve enough statistical power for sound binary hypothesis testing (GPT 4 vs 3.5 in the different tasks, additionally Google for diagnosis task). Including further models would result in many more pair-wise comparisons. This in turn would require a disproportionate amount of further new patient case reports that need to be evaluated for all models to ensure statistical robustness.

We therefore decided to remain our primary analysis on the statistical testing between GPT 3.5 and 4, but added a further descriptive analysis by comparing total scores between all four models – ChatGPT 3.5; 4; Llama-2-7b-chat; Llama-2-70b-chat - for a subset of 18 cases, which are balanced in terms of disease frequency and specialty. To do this we have deployed and configured two Llama 2 models that the reviewer has suggested and reevaluated these cases again with two independent physicians for the two new model configurations. We have included and referenced all configuration details. We believe the new results are highly interesting, as it shows the potential of open source models, yet maybe unknown in the medical community. We added a further section of discussion to underline the usage of such models for local deployment that not only fosters transparency or trustworthiness but also data privacy. We would like to express our special thanks for this specific comment, as it inspired us to further deploy and research open source LLMs at our university hospital as we were positively surprised by their base performances and the fact that they can more or less compete with ChatGPT and increase transparency while preserving data privacy.

- It is known that LLMs are sensitive to how the prompts are phrased, which is why we spend much time on prompt engineering to generate desired outputs stably. The authors mainly focus on an online situation where a consumer seeks medical consultation. Therefore, the input language is expected to be varied, with some prompts better at querying certain LLMs than others. The comparison between GPT-3.5 and GPT-4 in this paper specifies a common prompt template, which makes the comparison “fair” but not enough to reflect these LLMs’ capability to comprehend various ways of asking medical questions.

We consider the reviewer’s considerations on prompt engineering very valuable and have now expanded upon them in our revised discussion. Prompt engineering is a significant challenge in the study of LLMs. Optimized prompts frequently result in significantly improved outcomes, which we would also expect in the context of our topic. Nevertheless, we consider our methodology to be minimally biased, as we included ChatGPT via the public user interface. This allowed us to ensure that our results are in line to models currently in widespread usage. Furthermore, this approach is the only way to make a meaningful comparison with the Google search engine, which, like ChatGPT, is not specifically optimized for medical applications. In addition to the model site, we expect hardly many manual optimization of prompts from the normal user for whom our study is designed. Following this reasoning, conducting real user prompts would have been beneficial for our study. We agree with this, but it would have required markedly increased number of cases to remain statistical power for the evaluation of model outputs.

Other comments:

- Line 116: It is a good attempt to test GPT-3.5 and GPT-4 with information not seen in their training data. However, restricted data access cannot guarantee it. Tech companies competing in LLMs try every possible way to gain more training data, including scanning printed books and purchasing databases, legal or illegal. There have been multiple lawsuits alleging OpenAI using copyright materials to train GPT-3.5. I have also observed that GPT-3.5

and GPT-4 know very well how doctors write medical records in the Chinese language, which could only mean that they have seen Chinese medical records in their training data, and I could not think of a legal way how they obtained it in the US. Therefore, one should be very conservative when claiming GPT-3.5 or GPT-4 have not seen something in their training data.

Of course we cannot rule out that our original case reports were subject to training of OpenAI. As Reviewer 2 has pointed out (A1), we used our sources and preprocessing steps (reformulation into laymen-terms, translation to English) as additional safe guards. We have changed the wording in the introduction to: “... *the subsequent conversion into layman terms collectively serve as safeguards in order to minimize the likelihood that ChatGPT was trained on this input*”

- Line 160: The inclusion criterion rules out cases requiring imaging or lab results, which makes the problem much easier than real clinical encounters. A common criticism about automatic diagnosis models from doctors is that they overly rely on symptoms, while in practice, there are not many diseases that can be diagnosed without imaging and lab tests. The authors should at least mention this as a limitation.

We agree that ruling out imaging and lab tests is a limitation to diagnostic capabilities. However, our assessment focused on the initial diagnosis, where patients experience symptoms without having any imaging and lab-testing yet available. We have clarified this in the introduction, methods (subsection selecting clinical case reports) and discussion (subsection strengths and limitations). Moreover, we mention current possibilities to upload imaging data as part of LLM-conversation.

- Line 281: A p-value of 1 looks very strange. The lower part of Figure 2.A shows that the distribution of scores is different between GPT-3.5 and Google. Can the authors check if the test direction is correct in their one-sided test? They can swap the direction or use a two-sided test and see if the conclusion holds.

The reviewer is absolutely right with his comment. Although the direction of the test we applied was correct, we did – by error – adjust twice for multiple testing. The correct p value for the comparison of GPT-3.5 vs Google is 0.0518, the adjusted p value is thus 0.6125. We corrected this information in the main manuscript, Figure 2 and Table S2.

- Line 369: “As mentioned, the analyzed LLMs are continuously being updated through fine-tuning by their providers. This could lead to slightly different results if our test cases are re-entered, which limits the reproducibility of the exact performance results. Thus, our results need to be interpreted as snapshots of their time. To take this into account, we studied not only one LLM but two different successive versions with major updates (GPT-3.5 vs GPT-4).” The first part of this statement is true: these models are being updated by OpenAI, and the performance changes over time. Thus, one should experiment on different versions of the same model to study their change or stability, e.g., gpt-3.5-turbo-0613 vs. gpt-3.5-turbo-

0301, instead of two different models. Also, to improve reproducibility, the authors should specify the model version (e.g., gpt-3.5-turbo-0301) and use a temperature of 0.

We agree with the review's statement. It would be very useful to compare different versions to identify trends. This would have exceeded the scope of our study but should be investigated in future work.

With regard to exact versioning and parameters such as temperature and top_p, we would like to clarify that our requests were made through the user interface (chat.openai.com), as access to GPT-4 via the API was unavailable during our project timeline in April/May 2023. Therefore, we cannot specify the exact version and parameters. As we are aware that the models have adapted over time, we have specified the exact time of the queries. For all investigations based on Llama 2, we provide detailed information on the parameters and models.

- The language in the Introduction needs grammar editing and is not accurate enough. Starting from Methods, it feels much more fluent, accurate, and easy to follow.

We have re-written several sections of the introduction and removed some typos and grammar mistakes.

Many thanks for this review!

Reviewer #2

In terms of methodology:

A1. The study took good efforts to ensure test dataset was representative of realistic clinical cases and minimized bias. The dataset comprised case reports extracted from German clinical casebooks, across the major specialities in medicine (internal medicine, surgery, neurology, gynecology and pediatrics). The restricted access, unavailability in English, and subsequent conversion into layman terms, were safeguards taken by the authors to minimize the chance of ChatGPT being trained on the testing dataset. A decently sizeable number of cases (110 total cases) were included in the analysis.

A2. The workflow in extracting the final number of cases for the test dataset was clear and detailed. Of which, how was the random sampling of 40% of the total number of cases decided upon?

The reference to a detailed explanation of the 40% sample test was very valuable. We added more information on why we chose 40% to our methods section (subsection Selecting clinical case reports).

A3. Were there copyright issues breached in entering the content from non-open access clinical casebooks into GPT (which may be used in GPT model training)?

In order to avoid any copyright issues and promote the patient view, we have only used small fractions of the original German case reports (namely the medical history text), and have re-translated these text parts into English and further reformulated these into laymen text. Therefore, the remaining semantic content consists of commonly known patient characteristics of known disease entities but still contains new original word sequences, which were created by our manual curation.

A4. Good choice of statistical analyses were selected for the evaluation of responses.

Thank you very much.

In terms of results:

B1. For the cumulative frequency plots, it is less interpretable for the reader to look at the proportion of responses scored as each point on the Likert scale, for each tool.

We understand that the initial plots might not be intuitive enough. As an alternative representation of the distribution of scores for each tool, we added violin plots as Figure S5 to the Supplementary Information. The performance of each tool per task is visualized, distinguishing between rare, less frequent and frequent diseases. A reference was added to the results section of the main manuscript.

B2. More sub-analyses such as the pheochromocytoma case would be insightful, as there is a limit to the analysis of responses using the 5-point Likert scale based on 1 criteria only (clinical accuracy). Perhaps examples of 'dangerously erroneous' information which may cause harm to patients, if present, can also be included.

Sure, we have added one more exemplary case and discussed it very thoroughly. This case shows potential limitations, especially in the light of rare diseases and prompt engineering. Moreover, the reviewer also touches an important point regarding the our evaluation, using only overall clinical accuracy and ignoring potentially other dimensions. We have added this into our discussion and propose multi-dimensional clinical evaluations.

Many thanks for this review!

REVIEWERS' COMMENTS

Reviewer #1 (Remarks to the Author):

The authors have addressed my concerns.

Reviewer #2 (Remarks to the Author):

The review comments have been well rebutted and corrected by the authors.

Reviewer #2 (Remarks on code availability):

I am not able to find the code